# Enantioselective decarboxylative chlorination of β-ketocarboxylic acids

Kazutaka Shibatomi[1], Kazumasa Kitahara[1], Nozomi Sasaki[1], Yohei Kawasaki[1], Ikuhide Fujisawa[1] & Seiji Iwasa[1]

Stereoselective halogenation is a highly useful organic transformation for multistep syntheses because the resulting chiral organohalides can serve as precursors for various medicinally relevant derivatives. Even though decarboxylative halogenation of aliphatic carboxylic acids is a useful and fundamental synthetic method for the preparation of a variety of organohalides, an enantioselective version of this reaction has not been reported. Here we report a highly enantioselective decarboxylative chlorination of β-ketocarboxylic acids to obtain α-chloroketones under mild organocatalytic conditions. The present method is also applicable for the enantioselective synthesis of tertiary α-chloroketones. The conversions of the resulting α-chloroketones into α-aminoketones and α-thio-substituted ketones via $S_N2$ reactions at the tertiary carbon centres are also demonstrated. These results constitute an efficient approach for the synthesis of chiral organohalides and are expected to enhance the availability of enantiomerically enriched chiral compounds with heteroatom-substituted chiral stereogenic centres.

[1] Department of Environmental and Life Sciences, Toyohashi University of Technology, 1-1 Hibarigaoka, Tempaku-cho, Toyohashi 441-8580, Japan. Correspondence and requests for materials should be addressed to K.S. (email: shiba@ens.tut.ac.jp).

Organohalides are a fundamentally important class of compounds not only because the carbon-halogen bond is found in many biologically active compounds but also because a broad range of pharmaceutical and agrochemical agents are synthesized from organohalides, owing to the good leaving ability of halogen atoms[1–3]. Chiral halides are especially attractive building blocks because following the enantioselective introduction of the halogen atom onto a chiral carbon atom, subsequent substitution of the halogen atom via an $S_N2$ reaction, for example, generates a variety of chiral compounds with retained enantiopurity. Decarboxylative halogenation of aliphatic carboxylic acids is a useful organic transformation that directly replaces carboxyl groups with halogen atoms (Fig. 1a). Upon the initial discovery of this reaction by Borodine[4] and Hunsdiecker and Hunsdiecker[5], the transformation required the use of stoichiometric quantities of expensive transition metals. A number of improved methods, including the catalytic activation of carboxylic acids, have been developed over the past few decades[6–12]. For example, in 1997, Chowdhury and Roy[6] reported the first example of a catalytic decarboxylative halogenation reaction using lithium acetate as the catalyst, but this route was applicable only for cinnamic acid derivatives. In addition, in 2012, Li and colleagues developed a catalytic method for the decarboxylative chlorination and fluorination of general carboxylic acids. The visible-light-promoted decarboxylative halogenation has recently attracted much attention as a mild method to create carbon–halogen bonds[9–12]. In 2014, Sammis and Paquin initially revealed the visible-light-mediated decarboxylative fluorination of 2-aryloxyacetic acids in the presence of a Ru(II) photoredox catalyst[9]. Subsequently, MacMillan and colleagues succeeded in expanding the substrate scope of this reaction to a wide range of aliphatic carboxylic acids[10]. As these decarboxylative halogenation reactions can efficiently produce highly substituted alkyl halides, it was expected that an enantioselective version could be developed by employing a chiral catalyst in the reaction (Fig. 1b). However, to the best of our knowledge, this reaction has not been successfully adapted for asymmetric synthesis even though it was initially discovered over a century and a half ago. In this context, we considered the unique reactivity of β-oxocarboxylic acids, which have an additional carbonyl functionality at the β-position. As these carboxylic acids serve as an enolate equivalent involving the decarboxylation, mediation of decarboxylation step by a chiral catalyst is expected to result in reaction of the formal enolate with electrophiles in an enantioselective manner. Thus several groups have reported the enantioselective decarboxylative aldol-type reactions and 1,4-addition reactions of β-oxocarboxylic acids[13]. An elegant example has recently been reported by Saadi and Wennemers[14], in which decarboxylative aldol reactions of α-fluorinated malonic acid half thioesters and aldehydes yield the corresponding adducts in high diastereoselectivity and enantioselectivity in the presence of a chiral amine catalyst. Inspired by these previous studies, we envisaged that an enantioselective decarboxylative chlorination could be achieved by the chiral amine-mediated formation of enolates from racemic β-ketocarboxylic acids and subsequent electrophilic chlorination to yield enantiomerically enriched α-chloroketones via the ionic reaction pathway (Fig. 1c). This method would therefore be useful as an alternative to the direct α-chlorination of ketones for the synthesis of chiral α-chloroketones. Furthermore, although the development of methods for the enantioselective α-halogenation of carbonyl compounds has progressed rapidly in the past decade[15,16] since Hintermann and Togni[17] reported pioneering studies on the enantioselective halogenation of active methine compounds, only a single catalytic method has been reported for the enantioselective chlorination of unactivated ketones[18], and to

**Figure 1 | Decarboxylative halogenation of aliphatic carboxylic acids and development of an enantioselective reaction.** (**a**) The general reaction scheme of decarboxylative halogenation (R = unspecified alkyl group, Hal = unspecified halogen atom). (**b**) Desired enantioselective synthesis of non-racemic alkyl halides from racemic carboxylic acids. (**c**) Chiral amine-catalysed enantioselective decarboxylative chlorination of β-ketocarboxylic acids.

the best of our knowledge, no catalytic method is available for asymmetrical ketones[19]. Here we show a highly enantioselective decarboxylative chlorination of β-ketocarboxylic acids to obtain α-chloroketones with a chiral primary amine catalyst. $S_N2$ reactions of the resulting α-chloroketones at a tertiry carbon are also described[20,21].

## Results

**Reaction optimization.** Our synthetic study began with the preparation of β-ketocarboxylic acid **1a** as the starting material for the subsequent decarboxylative chlorination using known chiral amine catalysts, such as cinchona alkaloid derivatives and diarylprolinol derivatives (Fig. 2). However, although the desired α-chloroketone **2a** was obtained in good yield, the resulting enantioselectivity was poor. We also examined the reaction with a chiral Lewis acid catalyst prepared from a spiro pyridyl monooxazoline ligand **L1** and copper(II) trifluoromethanesulfonate because it is known to catalyse electrophilic halogenation of β-ketoesters with high enantioselectivity[22–24]. However, the decarboxylative chlorination of **1a** using this catalyst also proceeded with poor enantioselectivity. Recently, we have reported the synthesis of a 1,1′-binaphthyl-based chiral amino ester **C1** and successfully applied it for enantioselective fluorination of α-branched aldehydes[25]. Fortunately, the use of **C1** catalyst in the decarboxylative chlorination of **1a** to give **2a** was highly enantioselective. The use of similar amino esters with smaller substituents on the 3,3′ position of the binaphthyl backbone significantly decreased the enantioselectivity of the reaction. Following optimization of the reaction conditions, we found that the highest enantioselectivity was obtained using toluene as the solvent with N-chlorosuccinimide (NCS) as the chlorinating reagent (Supplementary Table 1, entries 1–4). The reaction was highly enantioselective even with a lower catalyst loading, even though the reaction rate significantly decreased when 5 or 2.5 mol% of the catalyst was used (Supplementary Table 1, entries 6 and 7). Furthermore, the reaction could be carried out with high enantioselectivity even in the presence of an additional aliphatic carboxylic acid (cyclohexane carboxylic acid)

**Figure 2 | Screening of chiral catalysts for enantioselective decarboxylative chlorination.** β-Ketocarboxylic acids were treated with NCS (3.0 equiv.) in the presence of chiral catalyst (30 mol%) and the mixture was stirred for 24 h in the dark, unless otherwise noted. The reactions with **C5** and **C6** were carried out for 3 h while the reaction with **C8** was carried out for 1 h. The reaction using the complex of Cu(OSO$_2$CF$_3$)$_2$ (10 mol%) and **L1** (12 mol%) (Cu(OSO$_2$CF$_3$)$_2$/**L1**) was performed in the presence of MS 4 A. In all cases, the yield of **2a** corresponds to the isolated yield of the purified compound, and the e.e. for each compound was determined by chiral high-performance liquid chromatography (Et = ethyl group, $t$Bu = $tert$-butyl group).

although the yield was slightly decreased (Supplementary Table 1, entry 8).

**Substrate scope**. With the optimized conditions in hand, a range of β-ketocarboxylic acids were subjected to enantioselective decarboxylative chlorination to yield the corresponding α-chloroketones in good-to-excellent enantioselectivities (Fig. 3). A particularly high asymmetric induction was observed in the reaction of tetralone-derived β-ketocarboxylic acids **1a–1e**. The reaction tolerated the presence of nitrile, allylic and benzylic moieties. Furthermore, this method was applicable not only to the synthesis of tertiary α-chloroketones but also to the preparation of secondary α-chloroketones. In the latter case, a preliminary experiment yielded significant amounts of the undesired α,α-dichloroketones **3** (Supplementary Table 2, entry 1), likely due to chlorination of the α-monoalkyl-β-ketocarboxylic acids and subsequent decarboxylative chlorination. However, generation of **3** could be suppressed to <5% yield through the slow addition of NCS during the reaction, to afford the desired product in good yield and high enantioselectivity (Supplementary Table 2, entry 3). Substituents on the benzoyl moiety of **1k** slightly diminished the enantioselectivity of the reaction and the cyclic α-monoalkyl-β-ketocarboxylic acid **1s** resulted in the corresponding product **2s** with moderate enantioselectivity. The absolute configurations of products **2a** and **2k** were confirmed to be $S$ and $R$, respectively, by single crystal X-ray analysis of **2a** and comparison with the specific rotation of a derivative of **2k** with the reported value (see Supplementary Table 3 and Supplementary Methods).

**Reaction mechanism**. As shown in Fig. 4, we propose that the reaction proceeds via the base-mediated decarboxylation and subsequent chlorination of the resulting enolate (or enol) intermediate **4** (Fig. 4a). As no protonated products were observed, it appeared that chlorination of the enolate proceeded more rapidly than the competitive protonation to yield simple ketones **5**. However, an alternative reaction mechanism could involve an initial decarboxylative protonation to afford the corresponding ketone **5**, followed by amine-catalysed chlorination to yield α-chloroketones via the formation of enamine intermediate **6**. To discard the latter mechanism, we confirmed that ketone **5a** did not react with NCS in the presence of **C1** (Fig. 4b). The reaction also failed upon the addition of 1 equiv. of cyclohexane carboxylic acid ($c$-Hex-CO$_2$H) as a Brønsted acid source. It was also found that the primary amine moiety of catalyst **C1** was chlorinated to give **C1′** in 68% yield during the course of the reaction. **C1′** could be isolated by silica gel column chromatography along with 24% of **C1** (Fig. 4c). Therefore, we confirmed whether or not **C1′** works as the chlorination reagent or catalyst in the reaction. The enantioselective decarboxylative chlorination of **1a** with **C1′** instead of NCS as the chlorinating reagent gave **2a** in only 5% yield with poor enantioselectivity (Fig. 4d). The reaction without either NCS or **C1′** afforded the protonated product **5a** in 78% yield with 64% enantiomeric excess (e.e.; Fig. 4d). These results suggest that NCS could chlorinate the enolate intermediate in the reaction. When the chlorinated amine **C1′** was used as the catalyst instead of **C1**, the reaction afforded **2a** with decreased enantioselectivity in lower yield (Fig. 4d). Although it was found that **C1′** is also an active catalyst in the reaction, it is assumed that the reaction was mainly

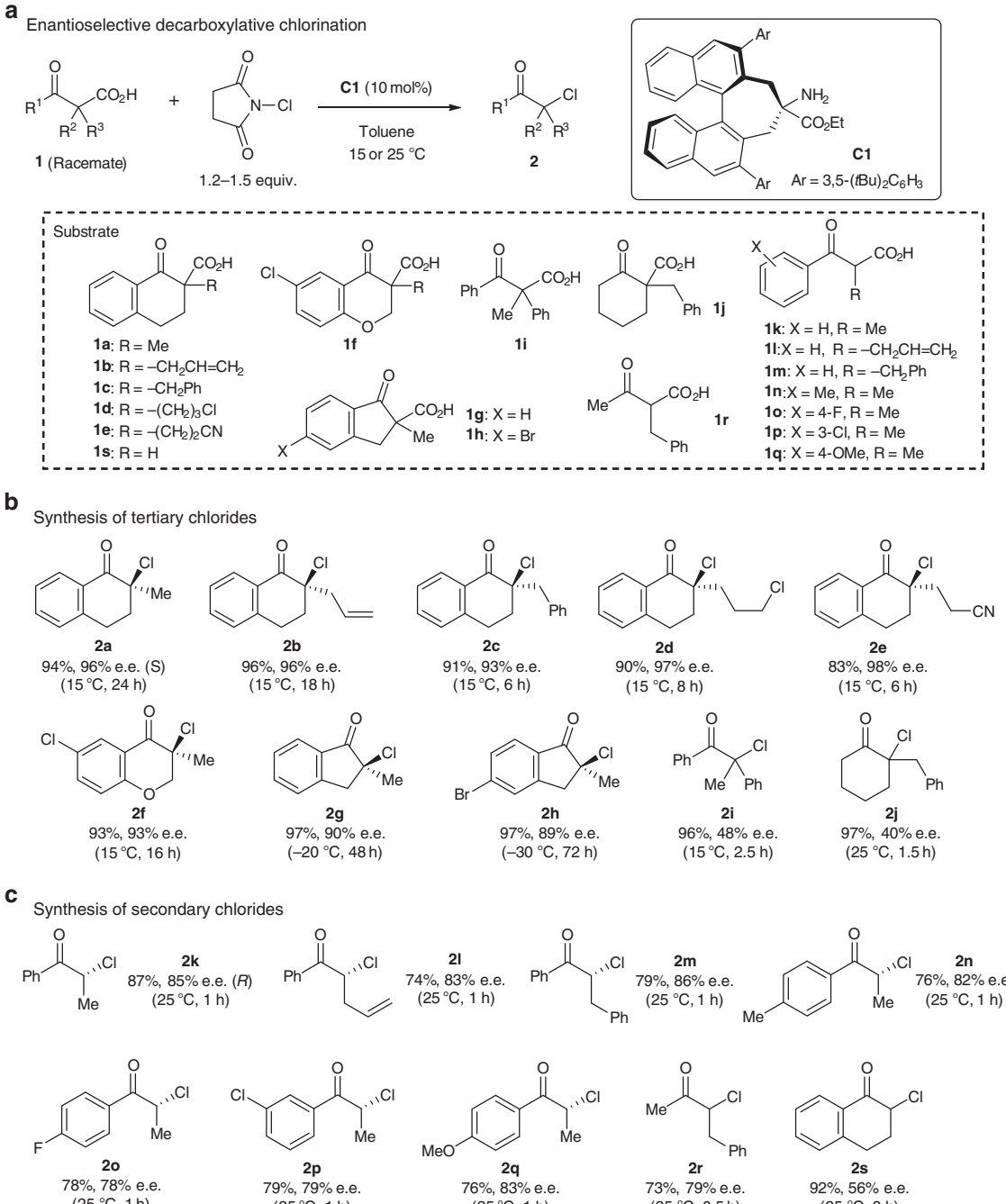

**Figure 3 | Scope of the catalytic enantioselective decarboxylative chlorination of β-ketocarboxylic acids. (a)** β-Ketocarboxylic acids were treated with NCS (1.2–1.5 equiv.) in the presence of catalyst **C1** (10 mol%) and the mixture was stirred according to the conditions indicated in the figure in the dark. In all cases, the yield of **2** corresponds to the isolated yield of the purified compound, and the e.e. for each compound was determined by chiral high-performance liquid chromatography. **(b)** Enantioselective decarboxylative chlorination of α,α-dialkyl-β-ketocarboxylic acids (Me = methyl group, Ph = phenyl group). **(c)** Enantioselective decarboxylative chlorination of α-monoalkyl-β-ketocarboxylic acids. During the reaction, a solution of NCS in toluene was added slowly to a stirred solution of **C1** and **1** in toluene over 1 h (0.5 h for the synthesis of **2r**) using a syringe pump.

catalysed by **C1** under the optimized conditions due to the lower concentration of **C1′** in the reaction mixture and its lower catalytic activity than **C1**. We also confirmed that the reaction of **5a** with **C1′** as the catalyst or chlorination reagent does not proceed at all (Fig. 4b).

**Derivatization of α-chloroketones.** We then explored the synthetic utility of these α-chloroketones **2** to demonstrate the

applicability of our novel process. We recently revealed that the S$_N$2 reaction of α-chloro-β-ketoesters proceeds smoothly to give the corresponding α-heteroatom-substituted-β-ketoesters with retained enantiopurity, despite the reaction taking place at a tertiary carbon atom[23,24]. Encouraged by this result, we examined the nucleophilic substitution of α-chloroketone **2a** (Fig. 5a). As expected, the reaction with NaN$_3$ proceeded smoothly to yield the corresponding α-azidoketone **7** in good yield without loss of enantiopurity. Compound **7** was then further

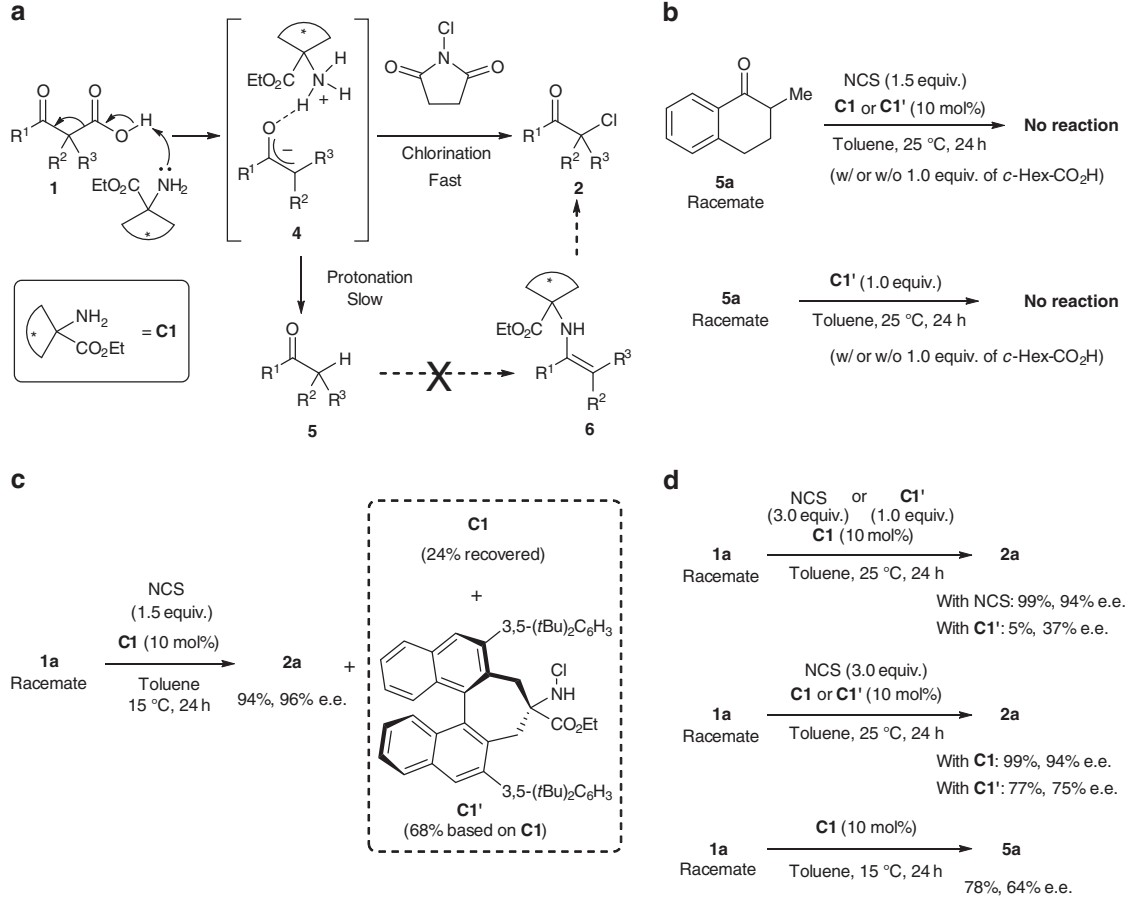

**Figure 4 | Proposed reaction pathway.** (**a**) A general mechanistic scheme for the decarboxylative chlorination of β-ketocarboxylic acids. (**b**) Neither **C1** nor **C1′** mediated the chlorination of ketone **5a** even in the presence of c-Hex-CO$_2$H (c-Hex = cyclohexyl). (**c**) **C1** was chlorinated during the course of decarboxylative chlorination to yield **C1′**. (**d**) Control experiments to identify the catalytic species and chlorination species.

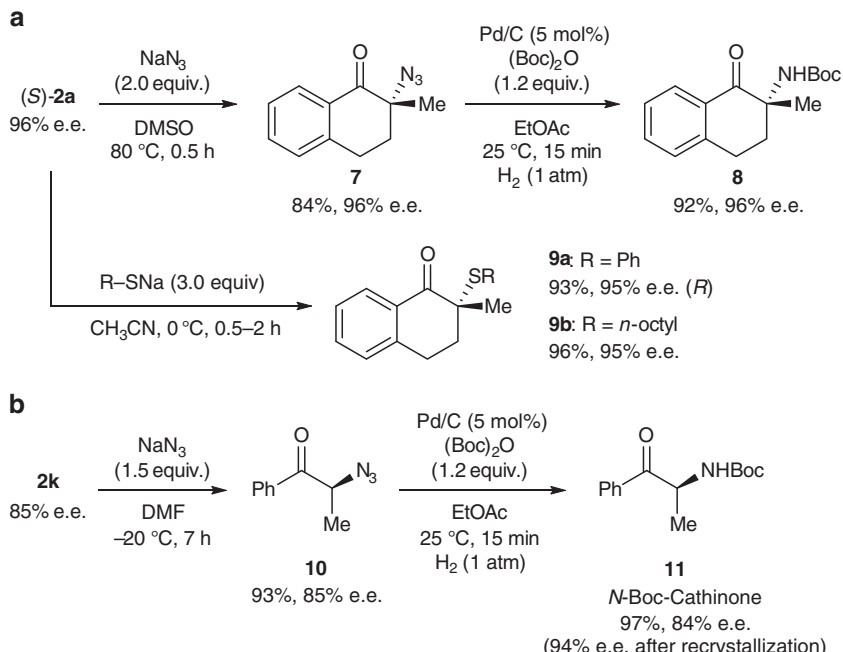

**Figure 5 | Stereospecific derivatization of α-chloroketones.** (**a**) S$_N$2 reaction of **2a** with either sodium azide or sodium thiolates as the nucleophiles (DMSO = dimethyl sulfoxide, EtOAc = ethyl acetate, Boc = tert-butoxycarbonyl group). (**b**) Enantioselective synthesis of the N-protected Cathinone (DMF = N,N-dimethylformamide).

converted into protected primary amine **8** under hydrogenolysis conditions. Similarly, **2a** was converted into α-sulfenylketones **9a** and **9b** in a stereospecific manner by treatment with the corresponding sodium thiolates. These smooth $S_N2$ reactions were assumed to be facilitated by a carbonyl function adjacent to the reaction centre[26]. We confirmed that the absolute configuration of **9a** was inverted from that of **2a** by single crystal X-ray analyses. This result strongly suggested that these nucleophilic substitutions proceeded in an $S_N2$ fashion with a Walden inversion. Although α-aminoketones and α-thio-substituted ketones are important molecules in the synthesis of a wide range of biologically relevant compounds, few catalytic methods for the enantioselective α-amination[27–29] or α-sulfenylation[30] of unactivated tertiary ketones are currently available. Our proposed method would constitute an efficient alternative to direct amination or sulfenylation procedures. Finally, we applied our novel method for the natural product synthesis (Fig. 5b). α-Chloroketone **2k** was initially converted into azide **10** by an $S_N2$ reaction with $NaN_3$. Subsequent hydrogenolysis of **10** accomplished the asymmetric synthesis of the *N*-carbamate-protected Cathinone **11** (ref. 31), which is a biologically active natural product that acts as a central nervous system stimulant. The enantiomeric purity of **11** was improved to be 94% e.e. after a single recrystallization.

## Discussion

This work describes the highly enantioselective decarboxylative halogenation of carboxylic acids. In this reaction, a chiral primary amine exhibiting an axial chiral 1,1′-binaphthyl backbone efficiently catalysed the decarboxylative chlorination of β-ketocarboxylic acids under mild organocatalytic conditions in high enantioselectivity. The resulting α-chloroketones acted as synthetic intermediates in the preparation of α-aminoketones and α-thio-substituted ketones via $S_N2$ reactions. Our method therefore constitutes an efficient approach to the simple preparation of multiple optically active compounds from a single intermediate. We therefore expect that this method will allow the synthesis of a wide variety of enantiomerically enriched α-heteroatom-substituted ketones and will be applicable in the preparation of novel pharmaceutical compounds. Studies into the application of this method to enantioselective fluorination reactions and to reactions with other types of aliphatic carboxylic acids bearing electron-withdrawing substituents are ongoing.

## Methods

**General.** All non-aqueous reactions were carried out in dried glassware under an argon atmosphere and stirred using magnetic stir-plates. Thin-layer chromatography analyses were performed using precoated silica gel plates with a fluorescent indicator (F254) (Merck Millipore, Darmstadt, Germany). Visualization was accomplished by ultraviolet light (254 nm), phosphomolybdic acid or *p*-anisaldehyde. Flash column chromatography was performed using silica gel 60 (mesh size 40–100) supplied by Kanto Chemical Co., Inc. (Tokyo, Japan). $^1H$, $^{13}C$ and $^{19}F$ NMR spectra were recorded on a JNM-ECS400 (400 MHz $^1H$, 100 MHz $^{13}C$, 376 MHz $^{19}F$) or a JNM-ECX500 (500 MHz $^1H$, 126 MHz $^{13}C$, 470 MHz $^{19}F$) instrument (JEOL Ltd., Tokyo, Japan). Chemical shift values (δ) are reported in parts per million (p.p.m.) (tetramethylsilane δ 0.00 p.p.m. or residual acetone δ 2.05 for $^1H$; hexafluorobenzene δ − 162.2 p.p.m. for $^{19}F$; residual chloroform δ 77.0 p.p.m. or acetone δ 29.8 p.p.m. for $^{13}C$). Infrared spectra were recorded on an FT/IR-4600 instrument (JASCO Co., Ltd., Tokyo, Japan). Direct analyses in real time mass (positive mode) analyses were performed on a JMS-T100TD time-of-flight mass spectrometer (JEOL Ltd.). Melting points were recorded on a YANACO MP-500 P micro melting point apparatus (Japan). Optical rotations were measured on a P-1030 digital polarimeter (JASCO Co., Ltd.). Analytical high-performance liquid chromatography (HPLC) was performed on a PU1586 instrument with a MD-2018 plus diode array detector (JASCO Co., Ltd.) using an appropriate chiral column. Complete experimental details can be found in Supplementary Methods.

**Materials.** Commercial grade reagents and solvents were used without further purification unless otherwise noted. Anhydrous acetonitrile, ethyl acetate,

*N,N*-dimethylformamide and dimethyl sulfoxide were purchased from Sigma-Aldrich (St Louis, MO). Anhydrous toluene, dichloromethane, tetrahydrofurane were purchased from Kanto Chemical Co., Inc. and used after purification by a Glass Contour solvent dispensing system (Pure Process Technology, Nashua, NH). The enantiomeric purity of the compounds was determined by HPLC analyses using chiral stationary-phase columns. β-Ketocarboxylic acids **1** were synthesized by acidolysis of the corresponding *tert*-butyl β-ketoesters. The catalyst **C1** was prepared by following a previously reported procedure[25].

**Decarboxylative chlorination of 1.** A typical procedure for the decarboxylative chlorination of tertiary carboxylic acids **1a–1j** outlined in Fig. 3 is as follows. To a stirred solution of catalyst **C1** (18.6 mg, 0.0245 mmol) and NCS (49.1 mg, 0.368 mmol) in toluene (1.2 ml) was added β-ketocarboxylic acid **1a** (50.1 mg, 0.245 mmol), and the reaction mixture was stirred at 15 °C for 24 h in the dark. After this time, the resulting mixture was subjected directly to silica gel column chromatography (hexane: dichloromethane = 2:1 to 1:2) to give the corresponding α-chloroketone **2a** as a white solid (44.9 mg, 94% yield, 96% e.e.). A typical procedure for the synthesis of secondary chlorides **2k–2 s** is as follows. To a stirred solution of catalyst **C1** (45.0 mg, 0.0594 mmol), NCS (7.9 mg, 0.059 mmol) and **1k** (105.8 mg, 0.594 mmol) in toluene (3.0 ml) was added slowly a solution of NCS (87.2 mg, 0.653 mmol) in toluene (9.0 ml) over 1 h using a syringe pump in the dark. Then the reaction mixture was stirred at 25 °C for another 5 min. The mixture was purified by flash column chromatography (hexane: dichloromethane = 2:1) on silica gel to give **2k** as a colourless oil (87.1 mg, 87% yield, 85% e.e.).

**Synthesis of 7.** $S_N2$ reaction of **2a** with $NaN_3$ was performed as follows (Fig. 5a). To a stirred solution of **2a** (122.7 mg, 0.63 mmol, 96% e.e.) in dimethyl sulfoxide (2.5 ml) was added $NaN_3$ (81.9 mg, 1.26 mmol), and the reaction mixture was stirred at 80 °C for 20 min. Diethyl ether and water were added to the mixture, and the mixture was washed with water. The combined organic layer was dried over anhydrous $Na_2SO_4$, concentrated and then purified by flash column chromatography on silica gel (hexane: ethyl acetate = 10:1) to give **7** as a colourless oil (106.7 mg, 84% yield, 96% e.e.).

**Synthesis of 8.** Hydrogenolysis of **7** was performed as follows (Fig. 5a). To a stirred suspension of Pd/C (10%, 2.9 mg, 0.027 mmol) in ethyl acetate (6.1 ml) was added a solution of **7** (110.3 mg, 0.548 mmol, 96% e.e.) and di-*tert*-butyl dicarbonate (143.6 mg, 0.658 mmol) in ethyl acetate (12.2 ml) under a hydrogen atmosphere. Then the reaction mixture was stirred at 25 °C for 15 min. The mixture was filtered, and the filtrate was concentrated and then purified by flash column chromatography on silica gel (hexane: ethyl acetate = 9:1) to give **8** as a white solid (139.4 mg, 92% yield, 96% e.e.).

**Synthesis of 9.** The general procedure for $S_N2$ reaction of **2a** with sodium thiolates is as follows (Fig. 5a). To a stirred suspension of NaH (60% in oil, washed with hexane, 36.5 mg, 1.52 mmol) in acetonitrile (2.0 ml) was added the corresponding thiols (1.52 mmol) at 0 °C, and the mixture was stirred at 0 °C for 1 h. Then a solution of **2a** (98.9 mg, 0.508 mmol, 96% e.e.) in acetonitrile (2.0 ml) was added, and the reaction mixture was stirred at 0 °C for 2 h. The reaction mixture was quenched by adding saturated $NH_4Cl$ aqueous solution at 0 °C and then extracted with diethyl ether. The combined organic layer was dried over anhydrous $Na_2SO_4$, concentrated and then purified by flash column chromatography on silica gel to give **9**.

**Synthesis of 10.** $S_N2$ reaction of **2k** with $NaN_3$ was performed as follows (Fig. 5b). To a stirred solution of **2k** (78.0 mg, 0.463 mmol, 85% e.e.) in *N,N*-dimethylformamide (1.9 ml) was added $NaN_3$ (45.1 mg, 0.694 mmol) at − 20 °C, and the reaction mixture was stirred at − 20 °C for 7 h. Diethyl ether and water were added to the mixture, and the mixture was washed with water. The combined organic layer was dried over anhydrous $Na_2SO_4$, concentrated and then purified by flash column chromatography on silica gel (hexane: ethyl acetate = 20:1) to give **10** as a colourless oil (75.5 mg, 93% yield, 85% e.e.).

**Synthesis of *N*-Boc-Cathinone (11).** Hydrogenolysis of **10** was performed as follows (Fig. 5b). To a suspension of Pd/C (10%, 2.3 mg, 0.022 mmol) in ethyl acetate (7.2 ml) was added a solution of **10** (75.5 mg, 0.431 mmol, 85% e.e.) and di-*tert*-butyl dicarbonate (112.9 mg, 0.517 mmol) in ethyl acetate (7.2 ml) under a hydrogen atmosphere. Then the reaction mixture was stirred at 25 °C for 15 min. The mixture was filtered, and the filtrate was concentrated and then purified by flash column chromatography on silica gel (hexane: ethyl acetate = 5:1) to give **11** as a white solid (103.8 mg, 97% yield, 84% e.e.). The resulting **11** was dissolved in a minimum amount of pentane and recrystallized at − 20 °C for 5 days. The crystal was filtered and the filtrate was concentrated to give 94% e.e. of **11** (23.7 mg, 82% yield).

**Data availability.** For [1]H and [13]C NMR spectra and the traces of HPLC analyses of compounds, please see Supplementary Figs 1–142. X-ray crystallographic data have been deposited in the Cambridge Crystallographic Data Centre (CCDC) database under accession codes CCDC 1516051 (**2a**) and CCDC 1516052 (**9a**). This data can be obtained free of charge from CCDC via http://www.ccdc.cam.ac.uk/. The ORTEP diagrams and crystallographic and structure refinement data for **2a** and **9a** are given in Supplementary Tables 3 and 4. All other data are available from the authors upon reasonable request.

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

## Acknowledgements

This work was supported by a Grant-in-Aid for Scientific Research on Innovative Areas 'Advanced Molecular Transformations by Organocatalysts,' (26105728) and a Grant-in-Aid for Challenging Exploratory Research (16K13993) from MEXT, Japan. Partial support from Daiichi Sankyo Co., Ltd. and Suzuki Memorial Foundation is also acknowledged. K.K. is grateful to the Leading Graduate School Program R03 of MEXT.

## Author contributions

K.K., N.S., Y.K. and I.F. performed the experiments and analysed the data. K.S. conceived and directed the project in occasional discussion with K.K., N.S., Y.K. and S.I. and also wrote the manuscript.

## Additional information

**Competing interests:** The authors declare no competing financial interests.

