## [Peer review file · Nature Communications]

Reviewers' comments:

Reviewer #1 (Remarks to the Author):

Rather than dealing with ordinary aliphatic carboxylic acids, the authors choose beta-keto acids as the substrates for enantioselective chlorodecarboxylation. The decarboxylation of beta-keto acids is pretty easy and follows a completely different mechanism, i.e., non-radical mechanism. As the authors also point out, chiral amine-catalyzed enantioselective decarboxylative aldol reactions of beta-keto acids has been reported (reference 17). The chlorination described in this manuscript is very similar to this literature work. It is not novel or significant enough to justify the publication in Nature Commun. It is more appropriate to be published in a more specialized journal such as J. Org. Chem. or Eur. J. Org. Chem.

Reviewer #2 (Remarks to the Author):

This manuscript describes a very interesting enantioselective chlorodecarboxylation of beta-ketocarboxylic acid derivatives. Overall, the manuscript is well-written and the results are clearly presented. The substrate scope is very good, affording the desired chlorides in excellent yields and good to excellent enantioselectivities. The novelty is also high, and thus the manuscript is at the level one would expect from a Nature Communications manuscript. However, there are some changes that need to be made prior to publication:

- 1) The standard for methodology papers is the substrate should be >0.25 mmol, with scales more typically ranging around 0.5 mmol. For Figure 2, all of the substrates are used in approximately 0.245 mmol, which is barely acceptable and likely results in higher errors in the isolated yields. The substrates in Figure 4 are used <0.15 mmol, which is far too small of an amount (especially as an example to demonstrate utility). The reactions in this figure should be run on a larger scale.
- 2) The proposed mechanism needs further experimental evidence. It is difficult to imagine that no protonation occurs when there is such a large pKa difference between the protonated catalyst and the enolate. An alternate proposal is that the catalyst reacts directly with NCS (the catalyst should be quantitatively chlorinated under the reaction conditions). The reaction enantioselectivity then derives from the chlorinated catalytic species. This mechanism is easy to test and the proper control experiments should be run prior to publication.
- 3) The authors refer to the transformation as a "Hunsdiecker-type" reaction. While the reaction described in this manuscript goes through a decarboxylation, a Hunsdiecker, or Hunsdiecker-type, reaction involves radical intermediates and does not use beta-ketoesters. As this process is purely ionic, the name is not applicable and should be changed.

Reviewer #3 (Remarks to the Author):

In this manuscript Shibatomi and co-workers report a new methodology for the enantioselective alpha-chlorination of ketones. The reaction makes use of a tailored amino-catalyst to trap ketone enolates generated via decarboxylation. The work represents a valuable step forward in asymmetric catalysis to prepare chiral halogenated building blocks useful for subsequent transformations. Importantly, the transformation can be used to prepare both secondary and tertiary substituted alpha-chloro ketones through minor modification of the procedure. While the sub-classes of ketones amendable to highly selective reactions are somewhat limited in the report (one R-group on the

ketone must be aromatic), continued optimization based on the concept presented should allow for a general approach. To the best of my knowledge the authors capture the state-of-the-art in the introductory paragraphs, this work represents the first enantioselective Hunsdiecker-type chlorination, and asymmetric halogenations to generate products corresponding to unactivated ketones remain limited (although accessible via two-step reactions).

Before publication of this excellent work, I suggest the authors consider the follow points.

1. Is the amino catalyst stable or chlorinated under the reaction conditions?
2. In the absence of NCS, one would expect non-productive decarboxylative to generate product 5, can the authors comment on what happens when substrate and catalyst are combined without NCS.
3. In the SI it is stated the reactions are performed in the dark, but this information is not in the extended data section. The use of a syringe pump is also noted in the SI but not in the extended data section. These entries should to be consistent. Specific reference to how the catalyst was prepared should be included (likely as in Ref 22). Otherwise, the experimental details are documented sufficiently to enable reproduction of the results.
4. It would be useful for the authors to examine what is the minimum catalyst loading required for reasonable yields and selectivity at extended reaction times (48 hours), especially given the high molecular weight and laborious synthetic route to the catalyst. Is the catalyst recoverable (by protonation/extraction) after the reaction?
5. Does the reaction tolerate unactivated alkyl carboxylic acids? This can be tested by the addition of *n*-Hex-CO₂H.
6. Can the authors comment on the results with the standard substrate when subjected to the reaction conditions reported in reference 23.
7. The comment in the introduction concerning radical intermediates: "The generation of such species renders it difficult to control the stereochemistry of the reaction." is not really accurate as many recently developed transformations suggest this is not true. The statement does not add to the discussion so it should be considered for removal.

Rylan Lundgren

Point-by-Point Response to Comments by Reviewers

Response to Reviewer #2:

I am very grateful for your valuable advice. The manuscript has been significantly revised based on your suggestions.

C1) The standard for methodology papers is the substrate should be >0.25 mmol, with scales more typically ranging around 0.5 mmol. For Figure 2, all of the substrates are used in approximately 0.245 mmol, which is barely acceptable and likely results in higher errors in the isolated yields. The substrates in Figure 4 are used <0.15 mmol, which is far too small of an amount (especially as an example to demonstrate utility). The reactions in this figure should be run on a larger scale.

R1) As per your suggestion, all reactions shown in Fig. 4 in the original manuscript were performed again on a scale greater than 0.4 mmol and these results are shown in Fig. 5 of the revised manuscript. Fortunately, the results were reproducible and the isolated yields of the compounds **7**, **9a**, **9b**, **10**, and **11** lay within the margin of error (at most 1% difference from the previously reported yields for reactions carried out on a smaller scale).

C2) The proposed mechanism needs further experimental evidence. It is difficult to imagine that no protonation occurs when there is such a large pKa difference between the protonated catalyst and the enolate. An alternate proposal is that the catalyst reacts directly with NCS (the catalyst should be quantitatively chlorinated under the reaction conditions). The reaction enantioselectivity then derives from the chlorinated catalytic species. This mechanism is easy to test and the proper control experiments should be run prior to publication.

R2) Following your suggestion, we found that the catalyst **C1** was chlorinated by NCS during the course of the decarboxylative chlorination. Indeed, the chlorinated catalyst **C1'** could be isolated after the reaction in 68% yield (based on **C1**). Therefore, we carried out the chlorination of β -ketocarboxylic acid **1a** with **C1'** as chlorinating reagent instead of NCS and found that **2a** was formed in only 5% yield with poor enantioselectivity. When **C1'** was used as a catalyst instead of **C1**, the reaction was less enantioselective and **2a** was obtained in a lower yield compared to the reaction where **C1** was used as the catalyst. On the basis of these results, we conclude that the chiral amine **C1** is the active catalytic species and NCS chlorinates the enolate intermediate. While I agree with you that protonation is likely to compete with chlorination in this reaction, we think that in this case the chlorination of enolate intermediate with NCS proceeds much faster than its protonation. All above-mentioned results have been included in the revised manuscript (main text line 111-122, Fig. 4c and 4d). These results support our proposed reaction mechanism.

C3) The authors refer to the transformation as a "Hunsdiecker-type" reaction. While the reaction described in this manuscript goes through a decarboxylation, a Hunsdiecker, or Hunsdiecker-type, reaction involves radical intermediates and does not use beta-ketoesters. As this process is purely ionic, the name is not applicable and should be changed.

R3) As per your suggestion, all mention of "Hunsdiecker (-type) reaction" in the manuscript has been changed to "decarboxylative halogenation".

Response to Reviewer #3:

I sincerely appreciate your constructive comments. We have revised the manuscript as follows:

C1) Is the amino catalyst stable or chlorinated under the reaction conditions?

R1) The amino catalyst **C1** is not stable per se under the reaction conditions and is chlorinated by NCS during the course of the decarboxylative chlorination. We found that the primary amine moiety of **C1** was chlorinated to yield the chlorinated amine **C1'** in 68% yield along with **C1** in 24% yield. These results have been described in the revised manuscript (main text line 111-113, Fig. 4c).

C2) In the absence of NCS, one would expect non-productive decarboxylative to generate product **5**, can the authors comment on what happens when substrate and catalyst are combined without NCS.

R2) In the absence of chlorinating reagent, the reaction affords protonated product **5** with moderate enantioselectivity. This result has been described in the revised manuscript (main text 116-117, Fig. 4d).

C3) In the SI it is stated the reactions are performed in the dark, but this information is not in the extended data section. The use of a syringe pump is also noted in the SI but not in the extended data section. These entries should be consistent. Specific reference to how the catalyst was prepared should be included (likely as in Ref 22). Otherwise, the experimental details are documented sufficiently to enable reproduction of the results.

R3) Based on your suggestions, the experimental procedure in the original manuscript has now been corrected. The statements explaining that decarboxylative chlorination was carried out in the dark and that the synthesis of secondary chlorides was performed with a syringe pump are now given in Fig. 2 and 3 in the revised manuscript. We have also described the method for preparing the catalyst in both Methods (line 184-185) and Supplementary Methods (page 53) and cited the reference literature.

C4). It would be useful for the authors to examine what is the minimum catalyst loading required for reasonable yields and selectivity at extended reaction times (48 hours), especially given the high molecular weight and laborious synthetic route to the catalyst. Is the catalyst recoverable (by protonation/extraction) after the reaction?

R4) The use of 5 mol% of **C1** results in the formation of **2a** in 97% yield with slightly decreased enantiomeric excess (95% e.e.) after 48 h. On the other hand, when 2.5 mol% of the catalyst was used, yield of **2a** decreased to 79% after 48 h with 94% e.e.. These results have been included in the revised manuscript (main text line 80-82 and Supplementary Table 1 (page 49), entries 6-7). As I mentioned above, catalyst **C1** was chlorinated during the course of decarboxylative chlorination to form **C1'**. While both **C1** and **C1'** could be isolated by silica gel column chromatography, they could not be extracted from the organic layer even after the protonation.

C5) Does the reaction tolerate unactivated alkyl carboxylic acids? This can be tested by the addition of *c*-Hex-CO₂H.

R5) We found that the reaction tolerated unactivated alkyl carboxylic acids. Following the reviewer's suggestion, the reaction of **1a** was carried out in the presence of cyclohexane carboxylic acid. As the result, **2a** was obtained in good yield with 95% e.e. even though the reaction rate was slightly decreased. This result has been described in the revised manuscript (main text line 82-84 and Supplementary Table 1 (page 49), entry 8).

C6) Can the authors comment on the results with the standard substrate when subjected to the reaction conditions reported in reference 23.

R6) Based on your comment, we examined the reaction of **1a** using the copper(II) complex of pyridyl oxazoline ligand **L1** as the Lewis acid catalyst. However, the reaction proceeded slowly to give **2a** with poor enantioselectivity. This result has been included in Fig. 2.

C7) The comment in the introduction concerning radical intermediates: "The generation of such species renders it difficult to control the stereochemistry of the reaction." is not really accurate as many recently developed transformations suggest this is not true. The statement does not add to the discussion so it should be considered for removal.

R7) We have deleted the statement indicated by the reviewer from the Introduction.

REVIEWERS' COMMENTS:

Reviewer #2 (Remarks to the Author):

In my original review I recommended publication after three changes to the manuscript. The first change was to increase the scale of some of the substrates. In the revised version of the manuscript, these reactions have been run on a larger scale. The second recommended change was to provide further experimental evidence for the proposed mechanism. In the newly revised version, further mechanistic experiments were described that address each of my comments/concerns. My final comment was minor and involved the use of the term "Hunsdiecker-type reaction". This has also been corrected in the revised manuscript. As all of my concerns have been appropriately addressed, I recommend publication without any further changes.

Reviewer #3 (Remarks to the Author):

The revisions improve the quality of the paper. The mechanistic findings are interesting, but not definitive, so some of the statements could be softened. See below for two specific comments in response to the initial comments and revisions.

C2) The proposed mechanism needs further experimental evidence. It is difficult to imagine that no protonation occurs when there is such a large pKa difference between the protonated catalyst and the enolate. An alternate proposal is that the catalyst reacts directly with NCS (the catalyst should be quantitatively chlorinated under the reaction conditions). The reaction enantioselectivity then derives from the chlorinated catalytic species. This mechanism is easy to test and the proper control experiments should be run prior to publication.

R2) Following your suggestion, we found that the catalyst C1 was chlorinated by NCS during the course of the decarboxylative chlorination. Indeed, the chlorinated catalyst C1' could be isolated after the reaction in 68% yield (based on C1). Therefore, we carried out the chlorination of β -ketocarboxylic acid 1a with C1' as chlorinating reagent instead of NCS and found that 2a was formed in only 5% yield with poor enantioselectivity. When C1' was used as a catalyst instead of C1, the reaction was less enantioselective and 2a was obtained in a lower yield compared to the reaction where C1 was used as the catalyst. On the basis of these results, we conclude that the chiral amine C1 is the active catalytic species and NCS chlorinates the enolate intermediate. While I agree with you that protonation is likely to compete with chlorination in this reaction, we think that in this case the chlorination of enolate intermediate with NCS proceeds much faster than its protonation. All above-mentioned results have been included in the revised manuscript (main text line 111-122, Fig. 4c and 4d). These results support our proposed reaction mechanism.

Comment: I suggest the authors be cautious with these statements. The control experiments are a bit perplexing to me; they do not provide a definitive answer. This is an interesting aspect of the work. The control experiments show that the chlorinated catalyst C1' is an active and selective catalyst in the reaction. It may be formed over a slower rate/exist in lower concentration than under the optimized conditions. This could explain the differences in reactivity between C1 and C1' in the presence of NCS. A final control experiment with 5a and C1' with and without NCS should be conducted.

C3) In the SI it is stated the reactions are performed in the dark, but this information is not in the extended data section. The use of a syringe pump is also noted in the SI but not in the extended data section. These entries should be consistent. Specific reference to how the catalyst was prepared should be included (likely as in Ref 22). Otherwise, the experimental details are documented

sufficiently to enable reproduction of the results.

R3) Based on your suggestions, the experimental procedure in the original manuscript has now been corrected. The statements explaining that decarboxylative chlorination was carried out in the dark and that the synthesis of secondary chlorides was performed with a syringe pump are now given in Fig. 2 and 3 in the revised manuscript. We have also described the method for preparing the catalyst in both Methods (line 184-185) and Supplementary Methods (page 53) and cited the reference literature.

Comment: Use of a syringe pump needs to be included in the methods section, ie Decarboxylative chlorination of 1 (Fig. 3).

Point-by-Point Response to Comments by Reviewer

Thank you again for your valuable comments on our manuscript. We have revised the manuscript as per your suggestions.

The revisions improve the quality of the paper. The mechanistic findings are interesting, but not definitive, so some of the statements could be softened. See below for two specific comments in response to the initial comments and revisions.

Comment: I suggest the authors be cautious with these statements. The control experiments are a bit perplexing to me; they do not provide a definitive answer. This is an interesting aspect of the work. The control experiments show that the chlorinated catalyst **C1'** is an active and selective catalyst in the reaction. It may be formed over a slower rate/exist in lower concentration than under the optimized conditions. This could explain the differences in reactivity between **C1** and **C1'** in the presence of NCS. A final control experiment with **5a** and **C1'** with and without NCS should be conducted.

As per the suggestion, statement describing catalytic active species in the reaction has been softened (page 5). We also carried out control experiments with **5a** and **C1'** and the results have been included in the revised manuscript (Fig. 4b). Thus, it was confirmed that neither **C1** nor **C1'** mediates the chlorination of **5a**.

Comment: Use of a syringe pump needs to be included in the methods section, ie Decarboxylative chlorination of **1** (Fig. 3).

Detailed experimental procedure with syringe pump for the synthesis secondary chloride (Fig. 3) has been described in Methods section (page 7).